# 6-Hydroxydopamine Induces Abnormal Iron Sequestration in BV2 Microglia by Activating Iron Regulatory Protein 1 and Inhibiting Hepcidin Release

**DOI:** 10.3390/biom12020266

**Published:** 2022-02-07

**Authors:** Manman Xu, Yinghui Li, Dapeng Meng, Danyang Zhang, Bingjing Wang, Junxia Xie, Jun Wang

**Affiliations:** 1School of Basic Medicine, Qingdao University, Qingdao 266071, China; xmm8908@sina.com.cn (M.X.); liyingbuhui@163.com (Y.L.); mdp911210@163.com (D.M.); zhangdy1996@163.com (D.Z.); wbj@163.com (B.W.); 2Institute of Brain Science and Disease, Shandong Provincial Key Laboratory of Pathogenesis and Prevention of Neurological Disorders, Qingdao University, Qingdao 266071, China; 3Medical Service Section, The Affiliated Hospital of Qingdao University, Qingdao 266071, China

**Keywords:** Parkinson’s disease, microglia, iron regulatory protein 1, hepcidin

## Abstract

Disrupted iron homeostasis in the substantia nigra pars compacta (SNpc) is an important pathological mechanism in Parkinson’s disease (PD). It is unclear what role microglia play in iron metabolism and selective iron deposition in the SNpc of PD brain. In this study, we observed that 6-hydroxydopamine (6-OHDA) induced the expression of divalent metal transporter-1 (DMT1) and iron influx in BV2 microglia cells, which might be associated with the upregulation of iron regulatory protein 1 (IRP1) expression. Moreover, we found that 6-OHDA had no significant effect on the expression of ferroportin 1 (FPN1) and iron efflux in BV2 microglial cells, which might be the combined action of IRP1 upregulation and reduced hepcidin levels. Furthermore, 6-OHDA treatment activated BV2 microglia and enhanced the release of pro-inflammatory cytokines. Interestingly, iron overloading suppressed IRP1 expression, thus downregulating DMT1 and upregulating FPN1 levels in these microglial cells. On the contrary, iron deficiency activated IRP1, leading to increased expression of DMT1 and decreased expression of FPN1—which indicates that activated IRP1 induces iron overloading in 6-OHDA-treated microglia, but not iron overloading modulates the expression of IRP1. Taken together, our data suggest that 6-OHDA can regulate the expression of DMT1 and FPN1 by activating IRP1 and inhibiting hepcidin release, thus leading to abnormal iron sequestration in microglia. In addition, 6-OHDA can activate microglia, which leads to increased release of pro-inflammatory factors that can further induce genome damage in dopaminergic neurons.

## 1. Introduction

Parkinson’s disease (PD) is the second most common motor neurodegenerative disorder, characterized by a progressive loss of dopaminergic neurons in the substantia nigra pars compacta (SNpc) [1,2,3]. The main clinical features of PD include bradykinesia, rigidity, rest tremor, and postural instability. Although heredity, environment, oxidative stress, and other etiological factors have been found to be responsible for the onset of PD pathology, the exact mechanisms by which these etiological factors induce PD pathogenesis have not been fully revealed [4,5].

Iron, an essential nutrient for almost all biological organisms, is widely involved in both physiological and biochemical functions [6,7,8]. Previous studies [9,10,11,12,13,14] have shown that the aggregation or sequestration of free iron may occur due to the abnormal expression of free iron carriers, such as divalent metal transporter 1 (DMT1) and ferroportin 1 (FPN1), which play a central role in PD pathogenesis. Iron regulatory proteins (IRPs) can translationally regulate iron transport proteins by binding to iron-responsive elements (IREs) presenting in the untranslated regions (UTR) of their respective mRNAs [15,16,17]. Binding of IRP to the IRE of the 5′-UTR of FPN1 inhibits FPN1 mRNA translation. Likewise, binding of IRP to the IRE of the 3′-UTR of DMT1 increases DMT1 mRNA translation [18,19,20]. Hepcidin is an iron-regulating hepatic peptide hormone discovered in recent years. It can inhibit the transcription and translation of the *FPN1* gene, promote the internalization and degradation of FPN1, reduce FPN1 levels, and play an important role in maintaining the balance of iron homeostasis [21,22]. Although recent studies have been primarily focused on iron deposition in dopaminergic neurons in PD, glial cells have also been shown to play key roles in maintaining iron homeostasis in PD [23,24]. Notably, the midbrain SNpc of PD patients has exhibited multifocal microglial activations which co-exist with a high amount of iron deposition [25]. However, the relationship between microglial activation and iron accumulation has not been completely elucidated. Our previous findings [9,10,11,12,13] have shown that 6-hydroxydopamine (6-OHDA) can activate IRP1, which in turn modulates the increased expression of DMT1, and the decrease in FPN1 expression as well—further leading to the accumulation of iron and the toxic injury of dopaminergic neurons. Moreover, it has been shown that astrocytes treated with 6-OHDA exhibit increased expression of both DMT1 and FPN1, along with their enhanced ability to transport free iron in these cells—thus preventing pathogenic iron deposition [26,27]. Based on these findings, we hypothesized that microglia might also play a crucial role in iron metabolism and region-specific iron deposition in the SNpc of the PD brain. In this study, we aimed to explore the effects of 6-OHDA treatment on BV2 microglial cell lines and related mechanisms to clarify the roles of microglia in selective iron deposition in the SNpc of the PD brain.

## 2. Materials and Methods

### 2.1. Reagents

6-OHDA, deferoxamine (DFO), ferric ammonium citrate (FAC), L-ascorbic acid, iron (II) sulphate heptahydrate (FeSO_4_·7H_2_O), dimethyl sulfoxide (DMSO), and anti-FPN1 antibody were obtained from Sigma (St. Louis, MO, USA). The anti-IRP1 antibody was obtained from Alpha Diagnostic International (ADI). The anti-DMT1 antibody was purchased from OriGene Technologies, Inc. Rabbit anti-β-actin antibody was purchased from Bioss. Hepcidin enzyme-linked immunosorbent assay (ELISA) kit was purchased from DRG International, Inc. We purchased Calcein-AM from Molecular Probes Inc. High glucose Dulbecco’s Modified Eagle’s Medium (DMEM) and heat-inactivated fetal bovine serum (FBS) were obtained from Gibco. All other chemicals and reagents were of the highest reagent grade available and were obtained from local commercial sources.

### 2.2. BV2 Microglial Cell Culture

BV2 microglial cells were procured from the Cell Bank of the Shanghai Institute of Cell Biology and Biochemistry, Chinese Academy of Sciences (Shanghai, China). They were grown in DMEM high glucose medium supplemented with 10% FBS, 100 U/mL penicillin, and 100 U/mL streptomycin (pH 7.4) in a humidified incubator containing 5% CO_2_ at 37 °C. For experiments, cells were seeded at a density of 1 × 10^5^/cm^2^ in 12-well plates and were grown to 80–90% confluency before they were treated with 10 μM of 6-OHDA or 100 μM FAC for 24 h, and then, cells were harvested for downstream experiments.

### 2.3. Drug Treatment

6-OHDA was dissolved in a solution containing 200 μg/mL of L-ascorbic acid in 0.9% NaCl to create a stock concentration of 1 mM. We used the final concentration of 10 μM for experiments. To investigate the effects of IRP1 activation on the expressions of DMT1 and FPN1, hepcidin release, and pro-inflammatory cytokine expression, the BV2 microglia were separated into either mock or 10 nM 6-OHDA-treated groups. For mock treatment, cells were treated with the medium alone and no 6-OHDA.

We dissolved 2.65 mg FAC in 1 mL basic medium to constitute a stock concentration of 10 mM. We used a final concentration of 100 μM for our experiments. To observe the expressions of IRP1, DMT1, and FPN1, BV2 microglial cells were divided into two groups: mock and 100 μM FAC-treated groups.

### 2.4. ELISA of Hepcidin

BV2 microglia were seeded in 12-well plates, pre-incubated for 12 h with 100 μmol/L FAC. Culture supernatants were collected, and the concentrations of hepcidin were determined using ELISA kits (DRG International, Inc., Edison, NJ, USA), as described in the manufacturers’ instructions.

### 2.5. Western Blotting

BV2 microglial cells were treated as described above. Following three washes with cold phosphate-buffered saline, cells were lysed with lysis buffer containing 50 mM Tris HCl pH 8.0, 150 mM NaCl, 1% Nonidet P-40, 0.5% sodium deoxycholate, 1 mM EDTA, 1 mM phenylmethylsulfonyl fluoride (PMSF), and protease inhibitors (pepstatin 1 μg/mL, aprotinin 1 μg/mL, leupeptin 1 μg/mL) for 30 min on ice. The insoluble pellet was removed by centrifugation (12,000 rpm, 20 min, 4 °C) from each sample. Protein concentration was determined by the Bradford assay kit (Bio-Rad Laboratories, Hercules, CA, USA). A total of 30 μg of protein was resolved by running in 8% SDS-polyacrylamide gels and was subsequently transferred to PVDF membranes. After blocking with 10% non-fat milk at room temperature for 2 h, the membranes were incubated with rabbit anti-rat IRP1 (1:800), DMT1 (1:800), and FPN1 (1:800) antibodies overnight at 4 °C. Anti-rabbit secondary antibody conjugated to horseradish peroxidase (Santa Cruz Biotechnology, Santa Cruz, CA, USA) was used at a 1:10,000 dilution. Blots were also probed with anti-β-actin antibody (1:10,000) as a loading control. Cross-reactivity was visualized using ECL western blotting detection reagents and analyzed by scanning densitometry using a UVP BioDoc-Imaging System (UVP, Upland, CA, USA).

### 2.6. Total RNA Extraction and Real-Time PCR Analysis

Total RNA was isolated using Trizol Reagent (Takara Biomedical Technology (Beijing) Co., Beijing, China) from BV2 microglial cells treated as described above, according to the manufacturer’s instructions. Then, 5 μg total RNA was reverse transcribed in a 20 μL reaction mixture with oligo-dT primers using a reverse-transcription system (Takara Biomedical Technology (Beijing) Co., Beijing, China). SYBR Green-based quantitative real time RCR (qRT-PCR) was employed to measure the relative changes in interleukin-1β (IL-1β) and tumor necrosis factor-α (TNF-α) expressions. The primers were designed and synthesized by (Takara Biomedical Technology (Beijing) Co., Beijing, China).

IL-1β forward: 5′-TCCAGGATGAGGACATGAGCAC-3′,reverse: 5′-GAACGTCACACACCAGCAGGTTA-3′.TNF-α forward: 5′-TATGGCCCAGACCCTCACA-3′,reverse: 5′-GGAGTAGACAAGGTACAACCCATC-3′.GAPDH gene was used as the reference gene:forward: 5′-AAATGGTGAAGGTCGGTGTGAAC-3′,reverse: 5′-CAACAATCTCCACTTTGCCACTG-3′.

Reactions were carried out in an Eppendorf Realplex4S System using the relative quantification option of the Mastercycler ep Realplex 2.2 software. Each reaction was run in triplicate, with 2 μL of each sample in a total reaction volume of 20 μL with primers to a final concentration of 0.25 μM. A passive reference dye, ROX II, which was not involved in the amplification, was used to correct the fluorescent fluctuations resulting from changes in the reaction conditions for normalization of the reporter signals. Amplification and detection were performed with the following conditions: an initial hold at 95 °C for 5 min, followed by 40 cycles at 95 °C for 5 s and 60 °C for 30 s.

### 2.7. Calcein Loading of Cells and Ferrous Iron Influx and Efflux Assay

The iron (II) trafficking in astrocytes was determined by measuring the quenching or reversal of calcein fluorescence, as previously described [23]. Briefly, cells were seeded onto coverslips and grown in serum-free medium with 10 μM 6-OHDA for 24 h. Then, they were incubated with calcein-AM (0.5 μM final concentrations of calcein-AM) in Hepes-buffered saline (HBS; 10 mM Hepes, 150 mM NaCl, pH 7.4) for 30 min at 37 °C. For iron influx into astrocytes, cells were perfused with 1 mM iron (II) salt (ferrous sulfate in ascorbic acid solution; 1:44 molar ratio, pH 6.0). Calcein fluorescence was recorded using an Olympus FV5 confocal microscope at 488 nm excitation and 525 nm emission wavelengths, and the corresponding fluorescence intensities were measured for 10 times at 3 min intervals.

For the iron (II) efflux assay, the coverslip was washed with HBS for three times after 30 min of incubation with 1 mM iron (II) solution, and then was exposed to 1 mM DFO—a membrane-impermeable, strong and specific iron chelator. Thus, the intracellular iron was drained out to the medium, leading to an increase in calcein fluorescence, which was recorded as described above. The fluorescence intensity was represented as the mean value of 35–40 cells from four different fields monitored at 20× magnification at each time point using Fluoview 5.0 Software (Leica Microsystems Inc., Buffalo Grove, IL, USA).

### 2.8. Statistical Analysis

Results were presented as mean ± SEM. Student’s two-tailed t-tests were used to compare the difference of means in experiments with only two groups. A one-way analysis of variance (ANOVA) followed by the Tukey test was used to compare differences of means in experiments with more than two groups. Influx and efflux studies were carried out using a repeated measure ANOVA. A probability of *p* < 0.05 was considered statistically significant.

## 3. Results

### 3.1. 6-OHDA Specifically Upregulates the Expression of DMT1 but Not FPN1 in BV2 Microglial Cells

First, we measured the expressions of DMT1 and FPN1 in 6-OHDA (10 μM, 24 h)-treated BV2 microglia cells by Western blotting. 6-OHDA could specifically induce the expression of DMT1 in microglial cells as compared with the control group (Figure 1A). The expression of FPN1 did not show any change in BV2 cells even after 10 μM of 6-OHDA treatment for 24 h, compared with the control group (Figure 1B). These results suggested that 6-OHDA can specifically and significantly upregulate the expression of DMT1 in BV2 microglial cells.

### 3.2. 6-OHDA Activates the Expression of IRP1 in BV2 Microglial Cells

To investigate whether IRP1 is activated in BV2 microglial cells treated with 6-OHDA, we performed Western blotting to measure the expression of IRP1. We found that the expression of IRP1 was increased in BV2 microglial cells treated with 10 μM of 6-OHDA for 24 h, as compared with the control group (Figure 2), suggesting that iron metabolism in BV2 microglial cells might be similar to that in neurons.

### 3.3. 6-OHDA Enhances Iron Influx in BV2 Microglial Cells without Affecting Iron Efflux

The fluorescent dye calcein was used to monitor iron (II) influx into primary BV2 microglial cells during 1 mM of iron (II) perfusion. The intracellular fluorescence intensity was gradually declined, indicating transmembrane iron (II) influx (Figure 3A). Next, we investigated the roles of microglial in regulating iron homeostasis under oxidative conditions. To do so, BV2 cells treated with 10 μM of 6-OHDA exhibited rapid fluorescence quenching and a corresponding decrease in fluorescence intensity compared with that of controls.

For the iron efflux assay, we treated BV2 cells with a potent iron chelator DFO. When cells were perfused with 1 mM of DFO, the fluorescence intensity did not show any change in the 10 μM of 6-OHDA-treated BV2 cells compared with that of controls (Figure 3B). These results indicate that 6-OHDA can enhance the iron influx capacity in BV2 microglial cells; however, it cannot affect the iron efflux capacity of the cell. These results are in accordance with the observed changes in iron-trafficking-related protein levels.

The ferrous trafficking in 6-OHDA-treated BV2 microglial cells was determined by the quenching and reversal of calcein fluorescence, an indicator of intracellular iron levels. The fluorescence intensity was calculated as the mean value of 35–40 cells from four different imaging fields at each time point and is presented as the mean ± SEM of six independent experiments (*n* = 6). (A) Perfusion of 1 mM of ferrous into BV2 cells showed a rapid decrease in fluorescence intensity in BV2 cells when treated with 6-OHDA as compared with the 6-OHDA untreated group, indicating increased iron influx due to 6-OHDA treatment (a repeated measure ANOVA, *F* = 25.96, *p* < 0.01, compared with controls). (B) A time-dependent reverse quenching with 1 mM of DFO perfusion in BV2 cells treated with 6-OHDA did not show rapid fluorescence reverse quenching compared with that in controls, indicating unchanged iron efflux in these cells (a repeated measures ANOVA, *p* > 0.05, compared with controls).

### 3.4. 6-OHDA Decreases Hepcidin Release in BV2 Cells

Given that the hepcidin–FPN1 axis regulates iron metabolism via the feedback effects of hepcidin, the hepcidin–FPN1 axis controls and regulates the body’s iron homeostasis [22,28,29]. To explore the regulatory mechanisms of FPN1 in 6-OHDA-induced BV2 microglial cells, we measured the amount of hepcidin release by ELISA. We found that hepcidin release was reduced in BV2 microglial cells upon treatment with 6-OHDA compared with that of the control group (Figure 4). This result suggests that 6-OHDA may decrease hepcidin release while inducing the expression of FPN1 in BV2 microglial cells.

### 3.5. 6-OHDA Increases the mRNA Expression of IL-1β and TNF-α in BV2 Cells

It has been well established that activated microglia release storms of pro-inflammatory cytokines, including IL-1β and TNF-α. Here, we found that the mRNA expression of IL-1β and TNF-α in 6-OHDA (10 μM, 24 h)-induced BV2 cells was significantly increased by qRT-PCR assay, compared with that of the control group (Figure 5). These results indicate that 6-OHDA treatment can induce the activation of IL-1β and TNF-α in BV2 microglial cells.

### 3.6. FAC Decreases the Expression of DMT1 and Increases FPN1 Expression in BV2 Cells

To investigate the roles of DMT1 and FPN1-mediated iron metabolism in iron overloaded microglia, we treated BV2 cells with 100 μM of FAC for 24 h. We found that the expression of DMT1 was significantly decreased in BV2 cells following FAC treatment, compared with that of the control group (Figure 6A). Meanwhile, the FPN1 expression was increased upon FAC treatment in BV2 cells, compared with that of the control group (Figure 6B). These results suggest that iron overloading in BV2 microglial cells may suppress DMT1 expression while increasing FPN1 levels to prevent abnormal iron sequestration.

### 3.7. FAC Decreases the Expression of IRP1 in BV2 Cells

To further explore the expression of IRP1 in iron overload conditions, we measured IRP1 protein levels by Western blotting in BV2 cells treated with 100 μM of FAC for 24 h or mocks. Our results showed that IRP1 levels were significantly reduced following FAC treatment (Figure 7), suggesting that IRP1 downregulation may enhance the expression of iron carrier proteins such as DMT1 in iron-overloaded microglia to prevent abnormal iron sequestration in these cells.

### 3.8. DFO Increases the Expression of DMT1 but Decreases FPN1 Expression in BV2 Cells

To investigate the mechanistic roles of iron carrier proteins in iron metabolism at reduced iron levels, we treated BV2 cells with 100 μM of DFO or mock treatment for 24 h. We observed that the expression of DMT1 was significantly increased in BV2 cells following the treatment, compared with the control group (Figure 8A). However, the expression of FPN1 was decreased in BV2 cells treated with DFO, compared with the control group (Figure 8B).

### 3.9. DFO Increases the Expression of IRP1 in BV2 Cells

To further explore the expression of IRP1 in low iron level conditions, we treated BV2 cells with 100 μM of DFO or mock treatment for 24 h. Then, we measured IRP1 protein levels by Western blotting, which showed a significant reduction in IRP1 levels following DFO treatment compared to the control group (Figure 9). This indicates that low iron levels in microglia upregulate the expression of IRP1 to regulate the expressions of DMT1 and FPN1, leading to an enhanced iron influx in microglia.

## 4. Discussion

This study suggests that 6-OHDA may induce the expression of DMT1 and iron influx in BV2 microglial cells, which may be associated with the upregulation of IRP1. 6-OHDA had no significant effect on the expression of FPN1, and iron efflux in BV2 microglial cells, which might be the result of IRP1 upregulation and hepcidin downregulation. Furthermore, 6-OHDA treatment showed microglia activation and subsequent release of inflammatory cytokines. Iron overload can suppress IRP1 expression—thus downregulating DMT1 expression and upregulating FPN1 levels. On the contrary, iron deficiency can activate IRP1 expression, leading to increased expression of DMT1, and decreased expression of FPN1. Thus, 6-OHDA can regulate the expression of DMT1 and FPN1 by activating IRP1 and inhibiting hepcidin release, leading to iron sequestration in microglia. In addition, 6-OHDA activates microglia, leading to the increased release of inflammatory factors, which may cause DNA damage in dopaminergic neurons.

According to the classical IRP/IRE system [9,10,11,12,13], the increased expression of IRP1 upregulates DMT1 levels and suppresses FPN1 expression. However, in BV2 microglial cells, we observed that the expression of FPN1 did not change, which suggests that there might be other regulatory factors regulating FPN1 levels and iron efflux in microglia.

Hepcidin is the main regulator of systemic iron homeostasis, doing so by binding to and causing the internalization and degradation of the iron exporter ferroportin [18]. Ferroportin degradation results in iron retention in cells; therefore, hepcidin increases intracellular iron stores and decreases serum iron levels [19,29]. It has been shown that hepcidin controls iron mobilization through its molecular target FPN1, in addition to IRP-mediated regulation. Moreover, hepcidin decreases FPN1 expression and vice versa [30]. In their latest study, Varga E et al. [29] indicated that inflammation mediated hepcidin regulation in microglia. Based on their results, it seems that both during inflammation and in normal conditions the absence of IL-6 triggers HAMP transcription and hepcidin secretion via the NF-κB pathway, and possibly via the autocrine effects of TNFα cytokine on BV2 microglia. In this study, using a hepcidin-based ELISA, we observed that 6-OHDA treatment decreased hepcidin release in BV2 microglial cells. This result might illuminate the unchanged expression of FPN1 in 6-OHDA-induced BV2 cells. This might be because of the dual regulation of FPN1 expression by IRP1 as well as hepcidin in microglia.

In this study, we observed that high iron levels (with FAC treatment) in BV2 cells decreased the expression of IRP1 and DMT1, while upregulating the expression of FPN1—leading to decreased iron influx and increased iron efflux, and thus preventing iron overload in microglia. However, under low iron levels (with DFO treatment), both the expression of IRP1 and DMT1 were increased, but FPN1 expression was downregulated—leading to increased iron influx and decreased iron efflux, to prevent iron deficiency in microglia. These results suggest that under normal physiological conditions, cellular iron levels modulate the expression of IRP1. We found that iron overload downregulated the expression of IRP1, and iron deficiency upregulated the expression of IRP1. Furthermore, our results showed that the expression of DMT1 was increased, while FPN1 expression remained unchanged in 6-OHDA-induced microglia, leading to enhanced iron influx and high intracellular iron levels. Although this might decrease the expression of IRP1 under iron overloaded conditions, the expression of IRP1 was increased. This indicates that intracellular iron levels might not affect the expression of IRP1 under 6-OHDA treatment, but might do so by activating IRP1 expression in microglia. Thus, it is activated IRP1 that can cause iron overloading in 6-OHDA-induced microglia, while not iron overloading can modulate the expression of IRP1.

Microglia are immune cells of the central nervous system and are implicated in brain inflammation related to both aging and neurodegenerative diseases [31]. It is widely accepted that microglial activation contributes to neurodegenerative disorders [32,33], such as PD [34], and Chen et al.’s [35] investigation has further fueled the idea that inflammation contributes to neurodegeneration. A study by Ryan C [31] has found that microglia preferentially take up non-transferrin-bound iron (NTBI) in response to pro-inflammatory stimuli such as lipopolysaccharides (LPS) or β-amyloid (Aβ), and that NTBI uptake is enhanced by the pro-inflammatory response; under these conditions, activated microglia sequesters both extra- and intra-cellular iron. In this study, we consistently observed that the expressions of TNF-α and IL-1β were increased in 6-OHDA-induced BV2 microglial cells. This indicates that 6-OHDA can activate BV2 microglial cells, which in turn enhances the release of pro-inflammatory cytokines (TNF-α and IL-1β)—thereby increasing iron influx and decreasing iron efflux in dopaminergic neurons, leading to iron deposition-induced DNA damage in these neurons.

## 5. Conclusions

6-OHDA can regulate the expression of DMT1 and FPN1 by activating IRP1 and inhibiting hepcidin, thus leading to iron sequestration in microglia. Furthermore, 6-OHDA can activate microglia, which leads to increased release of inflammatory cytokines, causing genome damage in dopaminergic neurons (Figure 10).

## Figures and Tables

**Figure 1 biomolecules-12-00266-f001:**
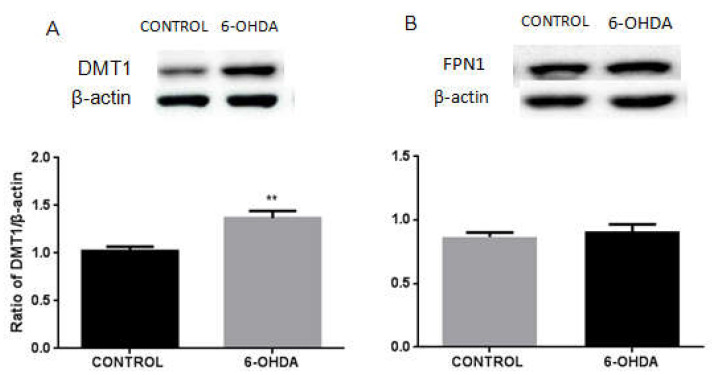
Protein levels of DMT1 and FPN1 in 6-OHDA-treated BV2 cells. (**A**) Western blot analysis to detect DMT1 protein levels. Increased expression of DMT1 was observed in 10 μM of 6-OHDA treated-BV2 microglial cells. β-actin was used as the loading control. Data are presented as the ratio of DMT1 to β-actin levels. (**B**) Western blot analysis to detect FPN1 protein levels. The expression of FPN1 did not change in 10 μM of 6-OHDA treated-BV2 microglial cells. β-actin was used as the loading control. Data are presented as the ratio of FPN1 to β-actin levels. Each bar represents the mean ± SEM from six independent experiments. *n* = 6. *** p* < 0.01, *p* > 0.05, compared with the control group.

**Figure 2 biomolecules-12-00266-f002:**
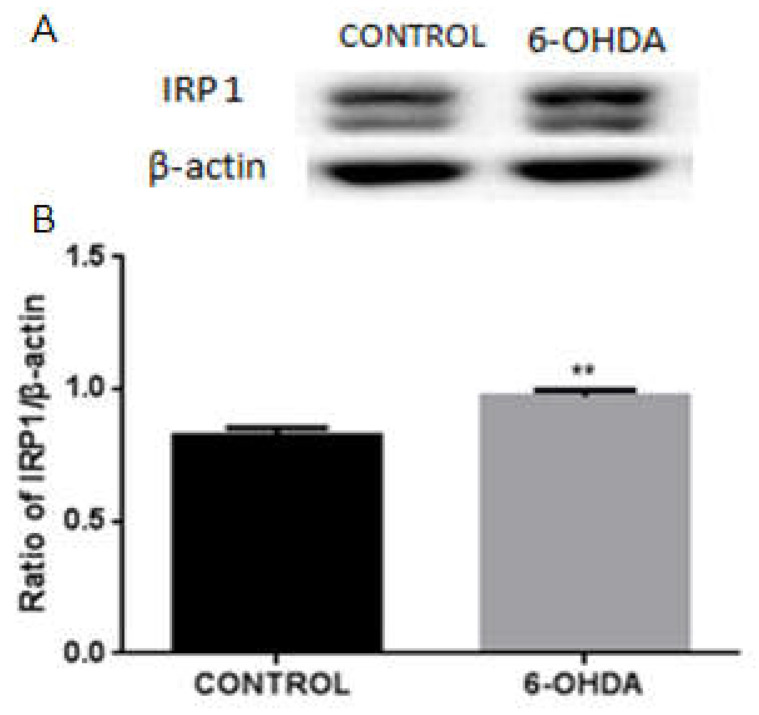
The protein levels of IRP1 in the 6-OHDA-treated BV2 cells. (**A**) Western blot analysis to detect IRP1 protein levels. Increased expression of IRP1 was observed in 10 μM of 6-OHDA treated BV2 microglial cells. β-actin was used as the loading control. (**B**) Data are presented as the ratio of IRP1 to β-actin levels. Each bar represents the mean ± SEM from six independent experiments. *n* = 6. ** *p* < 0.01, compared with the control group.

**Figure 3 biomolecules-12-00266-f003:**
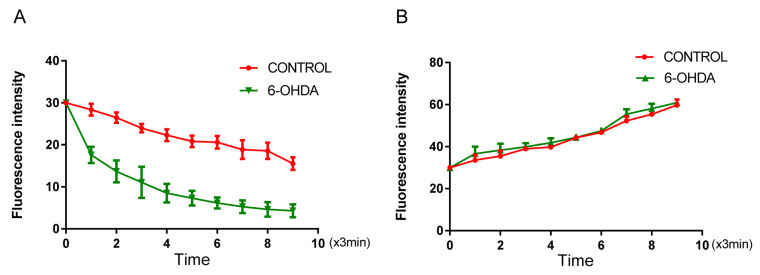
Modulation of iron trafficking in 6-OHDA-treated BV2 cells. (**A**) Calcein fluorescence was recorded at 488 nm excitation and 525 nm emission wavelengths and fluorescence intensity was measured every 3 min for 10 times while perfusing with 0.1 mM ferrous iron. For the iron efflux assay (**B**), primary cultured astrocytes were perfused with 1 mM ferrous iron for 20 min, then the fluorescence intensity was measured every 3 min for 10 times while perfusing with 1 mM DFO, a membrane-impermeant, strong and specific iron chelator, and the intracellular iron was drained out to the medium—indicated by the increase in calcein fluorescence. Data are presented as the mean ± SD of six independent experiments. *n* = 6.

**Figure 4 biomolecules-12-00266-f004:**
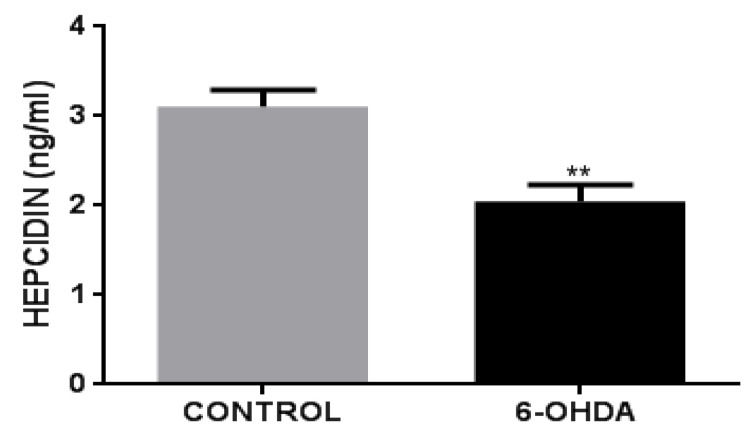
Amount of hepcidin release in 6-OHDA-induced BV2 cells. Hepcidin release was reduced in BV2 cells when treated with 6-OHDA compared with that of the control group. *n* = 6. ** *p* < 0.01, compared with the control group.

**Figure 5 biomolecules-12-00266-f005:**
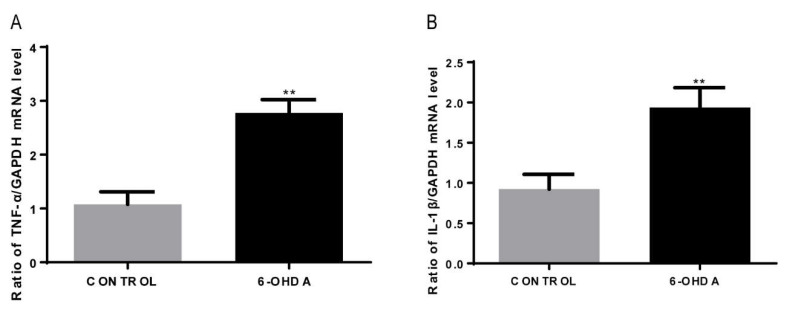
TNF-α and IL-1β mRNA levels in BV2 cells treated with 6-OHDA. (**A**) qRT-PCR assay to measure TNF-α mRNA levels. Data are presented as fold changes in TNF-α mRNA expression in 6-OHDA treatment versus control-treated groups. (**B**) qRT-PCR assay to measure IL-1β mRNA levels. Data are presented as fold changes in IL-1β mRNA expressions in 6-OHDA treatment versus control-treated groups. Each bar represents the mean ± SEM from four independent experiments. *n* = 4. ** *p* < 0.01, compared with the controls.

**Figure 6 biomolecules-12-00266-f006:**
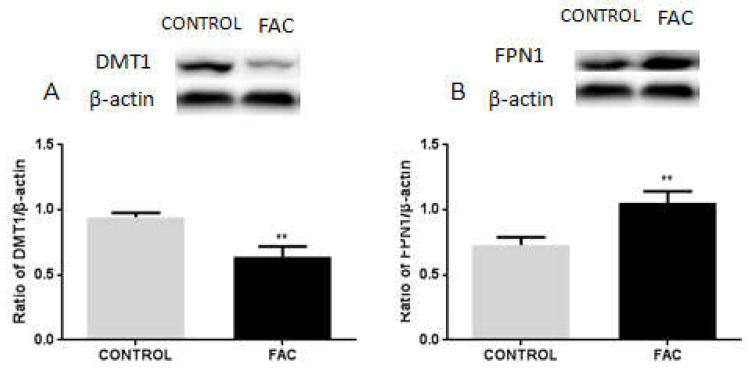
Protein levels of DMT1 and FPN1 in FAC or mock-treated BV2 cells. (**A**) Western blot shows a reduction in DMT1 levels in BV2 cells following 100 μM of FAC treatment in BV2 microglial cells compared to that of mock-treated cells. β-actin was used as the loading control. Data are presented as the ratio of DMT1 to β-actin levels. (**B**) Western blot reveals an increase in FPN1 expression in BV2 cells treated with 100 μM of FAC or mock treatment. β-actin was used as the loading control. Data are presented as the ratio of FPN1 to β-actin levels. Each bar represents the mean ±SEM from six independent experiments. *n* = 6. ** *p* < 0.01, compared with the control group.

**Figure 7 biomolecules-12-00266-f007:**
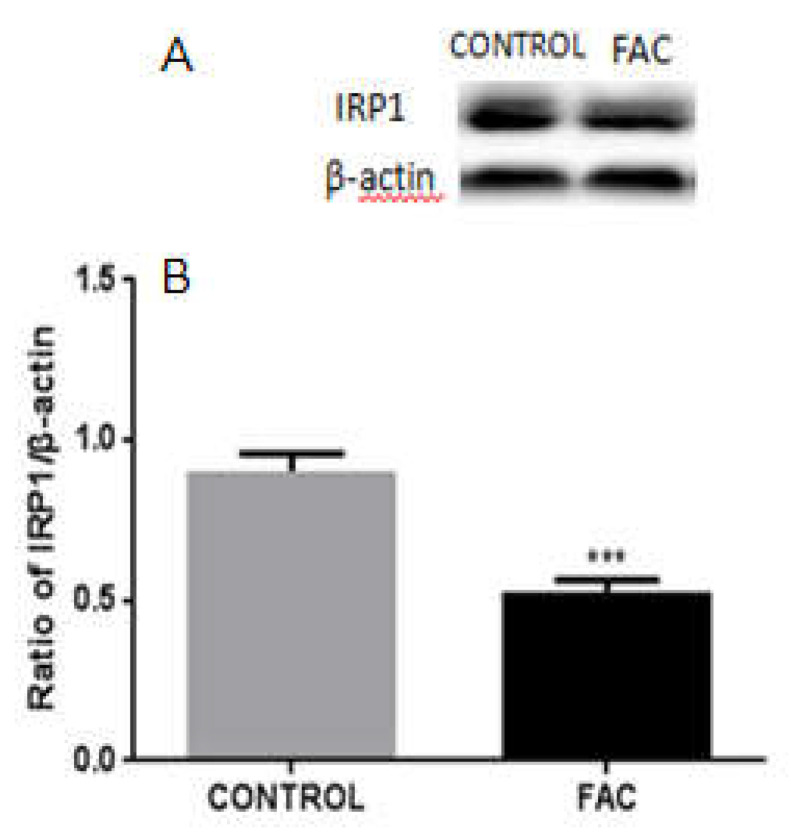
The protein levels of IRP1 in FAC-treated BV2 cells. (**A**) Western blot analysis of IRP1 protein levels. IRP1 levels were decreased in 100 μM of FAC-treated BV2 cells compared to those of mock-treated cells. β-actin was used as the loading control. (**B**) Data are presented as the ratio of IRP1 to β-actin levels. Each bar represents the mean ± SEM from six independent experiments. *n* = 6. *** *p* < 0.001, compared with the control group.

**Figure 8 biomolecules-12-00266-f008:**
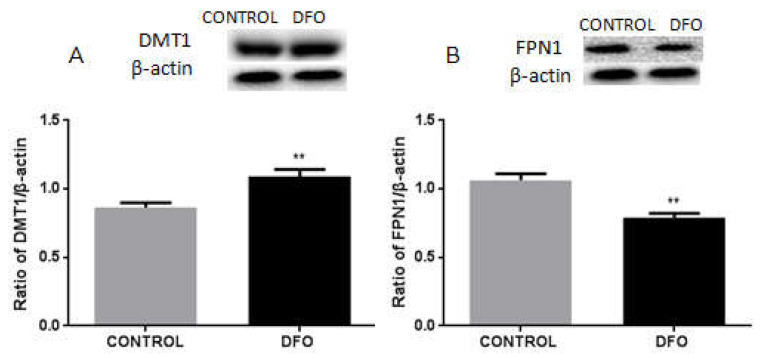
Protein levels of DMT1 and FPN1 DFO-treated BV2 cells. (**A**) Western blot analysis of DMT1 protein levels. DMT1 levels were increased in 100 μM of DFO treated-BV2 cells. β-actin was used as the loading control. Data are presented as the ratio of DMT1 to β-actin levels. (**B**) Western blot analysis of FPN1 protein levels. FPN1 levels were decreased in 100 μM of DFO treated-BV2 cells. β-actin was used as the loading control. Data are presented as the ratio of FPN1 to β-actin levels. Each bar represents the mean ± SEM from six independent experiments. *n* = 6. ** *p* < 0.01, compared with the control group.

**Figure 9 biomolecules-12-00266-f009:**
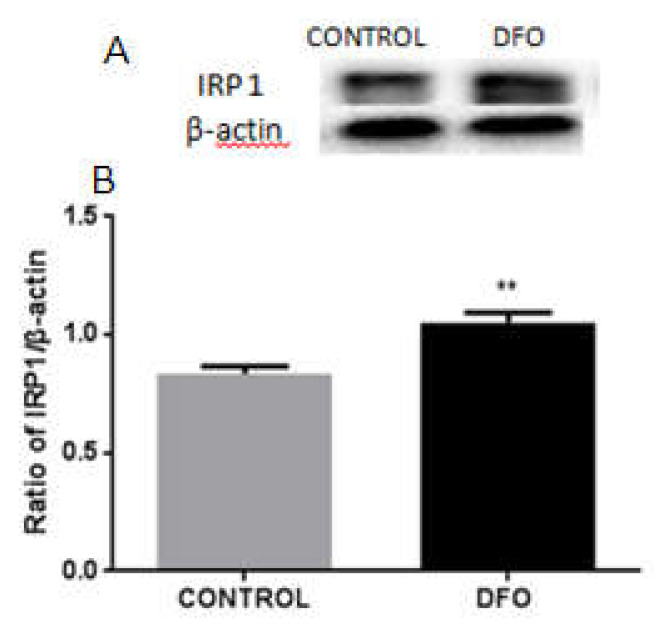
Protein levels of IRP1 in DFO-treated BV2 cells. (**A**) Western blot analysis of IRP1 protein levels. The expression of IRP1 increased in 100 μM of DFO treated-BV2 cells. β-actin was used as the loading control. (**B**) Data are presented as the ratio of IRP1 to β-actin level. Each bar represents the mean ± SEM from six independent experiments (*n* = 6), ** *p* < 0.001.

**Figure 10 biomolecules-12-00266-f010:**
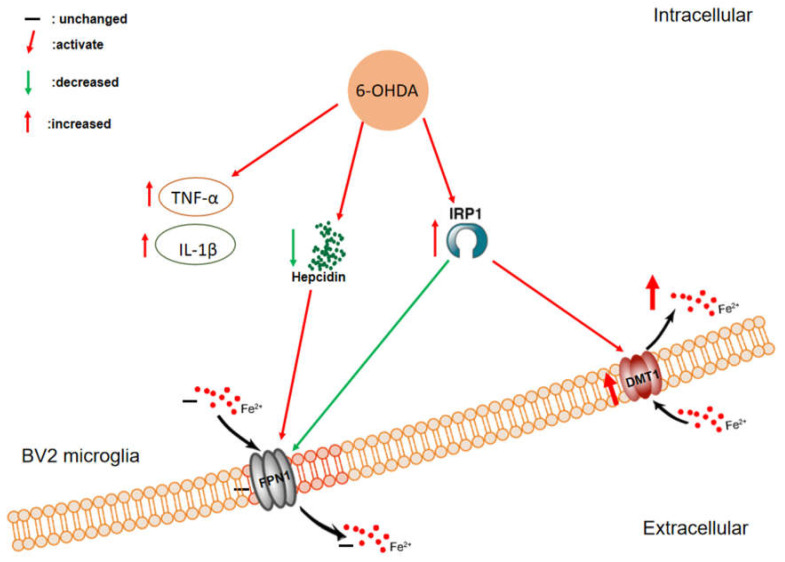
Graphical illustration of the effect of 6-OHDA on iron metabolism in BV2 cells. DMT1: divalent metal transporter 1; FPN1: ferroportin 1; IRP1: iron regulatory protein 1; IL-1β: Interleukin-1β; TNF-α: tumor necrosis factor-α.

## Data Availability

All data analyzed in this study are included within the article.

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
