# Peer review of "6-Hydroxydopamine Induces Abnormal Iron Sequestration in BV2 Microglia by Activating Iron Regulatory Protein 1 and Inhibiting Hepcidin Release"

_biomolecules, 2022, doi:10.3390/biom12020266_

Round 1

Reviewer 1 Report

This is a study which investigates an interesting concept regarding how an increased release of inflammatory cytokines by microglia can cause genome damage in dopaminergic neurons. Yet, the study has several experimental points that must be addressed both experimentally and by computational analysis. It is absolutely important that the quantification and statistical analysis reflect the real biological scenario of the outcome of the experiments. A technical limitation of this study is that all experiments were performed using BV2 microglia which are immortalized cells, some key experiments should also be performed using primary microglia.

Figure 1A: It is unlikely that the p value between CONTROL and 6-OHDA is so small (**) since the difference is 1.0 and less than 1.5 (ratio values).

Figure 2: Can authors explain the detection of a double band for IRP1? What band has been selected for the quantification?

Figure 3: has any statistical analysis been performed to compare the differences of fluorescence intensities between CONTROL and 6-OHDA?

Figure 4: same comment as above for Figure 1A regarding the statistical analysis. Why for hepcidin measurement ELISA has been used, while for the detection of other proteins western blot was performed?

Figure 5: mRNA expression levels should be accompanied by protein expression level analysis to confirm that protein expression reflects gene (mRNA) expression.

Figure 6: (A) the band intensity of DMT1 in FAC seems much less than half than CONTROL, this quantification should be repeated. Regarding the statistical analysis, same comment as for Figures 4 and 1A.

Figure 7: with a difference of less than 50% in the ratio of IRP1/beta-actin between CONTROL and FAC groups, the p value is even smaller (***). Statistical analysis should be repeated.

Figure 8: western blot analysis shows no important differences in DMT1 and FPN1 expression between CONTROL and DFO groups, therefore quantification and statistical analysis should be repeated more accurately.

Figure 9: same comment as Figures 6, 4 and 1A regarding the statistical analysis.

Author Response

Response to reviewer#1

Thank you for your insightful comments. We have made several corrections and changes to the manuscript based on your suggestions, which truly improved the quality of our work. Moreover, you have given us some good ideas for future work. Thank you once again.

  1. Q: Figure 1A: It is unlikely that the p value between CONTROL and 6-OHDA is so small (**) since the difference is 1.0 and less than 1.5 (ratio values). Figure 6: (A) the band intensity of DMT1 in FAC seems much less than half than CONTROL, this quantification should be repeated. Regarding the statistical analysis, same comment as for Figures 4 and 1A. Figure 7: with a difference of less than 50% in the ratio of IRP1/beta-actin between CONTROL and FAC groups, the p value is even smaller (***). Statistical analysis should be repeated. Figure 8: western blot analysis shows no important differences in DMT1 and FPN1 expression between CONTROL and DFO groups, therefore quantification and statistical analysis should be repeated more accurately. Figure 9: same comment as Figures 6, 4 and 1A regarding the statistical analysis.

A: Thank you for your suggestions. We have checked our data and results many times, as the same time, we also repeated those experiments. However, the p value is smaller as before. We discussed that, there might be two reasons: 1) The dispersion of data between independent duplicate groups is very small; 2) We used LSD tests, which is a highly sensitive method.

  1. Q: Figure 2: Can authors explain the detection of a double band for IRP1? What band has been selected for the quantification?

A: Thank you for your suggestion. Since 1997, Schalinske KL et. al.[1] demonstrated that IRP1 has two active forms Holo-IRP1 and Apo-IRP1, lots of works[2,3,4,5] also supported this finding. The references are listed below.

REFERENCE

  • Schalinske KL et. al. The iron-sulfur cluster of iron regulatory protein 1 modulates the accessibility of RNA binding and phosphorylation sites. 1997 Apr 1;36(13):3950-8.
  • Outten FW, Theil EC. Iron-based redox switches in biology. Antioxid Redox Signal. 2009 May;11(5):1029-46.
  • Soum E et. al. Peroxynitrite and nitric oxide differently target the iron-sulfur cluster and amino acid residues of human iron regulatory protein 1. 2003 Jul 1;42(25):7648-54.
  • Mehta M, Singh A. Mycobacterium tuberculosis WhiB3 maintains redox homeostasis and survival in response to reactive oxygen and nitrogen species. Free Radic Biol Med. 2019 Feb 1;131:50-58.
  • Urrutia PJ, Bórquez DA, Núñez MT. Inflaming the Brain with Iron. Inflaming the Brain with Iron. Antioxidants (Basel). 2021 Jan 6;10(1):61.

  1. Q: Figure 3: has any statistical analysis been performed to compare the differences of fluorescence intensities between CONTROL and 6-OHDA?

 A: Thank you for your question. In 2. MATERIALS AND METHODS part, 2.8. Statistical Analysis has showed that, Influx and efflux studies were carried out using a repeated measure ANOVA.

  1. Q: Figure 4: same comment as above for Figure 1A regarding the statistical analysis. Why for hepcidin measurement ELISA has been used, while for the detection of other proteins western blot was performed?

 A: Thank you for your question. There are two reasons why we used ELISA instead of western blot: 1) The molecular weight of hepcidin only 9 kDa, it’s too small to be detected by western blot; 2) Hepcidin is a protein that can be secreted out of the cells. We used R&D SYSTERM ELISA Kit (DHP 250), the Quantikine Hepcidin Immunoassay is a 4.5 hour solid-phase ELISA designed to measure Hepcidin in cell culture supernates, serum, plasma, and urine. It contains synthetic Hepcidin and antibodies raised against synthetic Hepcidin. This immunoassay has been shown to accurately quantitate synthetic and naturally occurring human Hepcidin.

Above all, compared with western blot, ELISA shows more efficient and accurate.

  1. Q: 6.Figure 5: mRNA expression levels should be accompanied by protein expression level analysis to confirm that protein expression reflects gene (mRNA) expression.

 A: We apologize for the lack of mRNA expression in this manuscript and thank you for your question. However, this study aimed to explore the protein functions and related effects of these iron related proteins. That’s the reason we didn’t observe the mRNA expression.

Reviewer 2 Report

Review of a manuscript “6-hydroxydopamine induces abnormal iron sequestration in BV2 microglia by activating iron regulatory protein1 and inhib[1]iting hepcidin release” by Manman Xu and coauthors submitted to “Biomolecules”

Parkinson’s disease is a severe neurodegenerative disorders, for which there is no treatment affecting its progression and no biomarkers for its early diagnostic. Iron plays an important role in Parkinson’s disease together with iron carriers, such as divalent metal transporter 1 (DMT1) and ferroportin 1 (FPN1).  The authors examined the effects of 6-hydroxydopamine (6-OHDA) treatment on BV2  microglial cell lines to elucidate a role of microglia in the selective iron deposition in the brain region substantia nigra pars compacta. This is important area of biomedical investigations and the results presented in the manuscript will be interesting for the readers of “Biomolecules”. The following corrections and additions should be made:

Abstract

The first eight lines describing previous author’s results are too detailed for the abstract. They can be transferred to Introduction or Discussion, but shortened or deleted from the Abstract.

Introduction

“Parkinson’s disease (PD) is the second most common motor neurodegenerative dis[1]order characterized by a progressive loss of dopaminergic neurons in the substantia nigra pars compacta (SNpc) [1].” The authors should add here a reference to a recent review on Parkinson’s disease: A Surguchov. Biomarkers in Parkinson’s Disease. pages 155-180, Jan 2022. In book: “Neurodegenerative Diseases Biomarkers.” Eds: PV Peplow, B Martinez, TA. Gennarelli, Springer Nature.

The sentence “When IRP binds to the IRE of the 5'-UTR of FPN1, FPN1 mRNA cannot be translated”  is awkward, it should be should be rewritten as follows: ”Binding of IRP to the IRE of the 5'-UTR of FPN1 inhibits FPN1 mRNA translation.”

The sense of the sentence “Likewise, when IRP binds to IRE located in the 3'-UTR of DMT1 mRNA, it stabilizes DMT1’s expression [13-15]” is unclear. What the authors mean by  “stabilizes DMT1’s expression”.

Figure 3 is of low quality and should be improved.

3.4 “Given that hepcidin-FPN1 axis regulates the iron metabolism by feedback effect of hepcidin (ref). : The sentence is unfinished. The references should be added.

Figure 5. The text on this figure is blurred, the quality of the figure should be improved.

Figure 10. The text at the left upper part is too small, it should be increased. Hepcidin should be placed on a place where it can be seen without interfering with other figure fragments.

Author Response

Response to reviewer#2

Thank you for your insightful comments. We have made several corrections and changes to the manuscript based on your suggestions, which truly improved the quality of our work. Moreover, you have given us some good ideas for future work. Thank you once again.

  1. Q: Abstract: The first eight lines describing previous author’s results are too detailed for the abstract. They can be transferred to Introduction or Discussion, but shortened or deleted from the Abstract.

A: Thank you for your suggestions. We have shortened abstract of the manuscript in red written.

  1. Q: Introduction:“Parkinson’s disease (PD) is the second most common motor neurodegenerative disorder characterized by a progressive loss of dopaminergic neurons in the substantia nigra pars compacta (SNpc) [1].” The authors should add here a reference to a recent review on Parkinson’s disease: A Surguchov. Biomarkers in Parkinson’s Disease. pages 155-180, Jan 2022. In book: “Neurodegenerative Diseases Biomarkers.” Eds: PV Peplow, B Martinez, TA. Gennarelli, Springer Nature.

A: We apologize for the lack of recent reviews and thank you for your suggestion. We have added the reference in red written.

  1. Q: The sentence “When IRP binds to the IRE of the 5'-UTR of FPN1, FPN1 mRNA cannot be translated”is awkward, it should be should be rewritten as follows: ”Binding of IRP to the IRE of the 5'-UTR of FPN1 inhibits FPN1 mRNA translation.”The sense of the sentence “Likewise, when IRP binds to IRE located in the 3'-UTR of DMT1 mRNA, it stabilizes DMT1’s expression [13-15]” is unclear. What the authors mean by “stabilizes DMT1’s expression”.

A: We apologize for this and thank you for your suggestion. We have rewritten in red written.

  1. Q: Figure 3 is of low quality and should be improved.

A: We apologize for this and thank you for your suggestion. We have replaced a new figure.

Fig 3.  Modulation of iron trafficking in 6-OHDA-treated BV2 cells.

  • Calcein fluorescence was recorded at 488 nm excitation and 525 nm emission wavelengths and fluorescence intensity was measured every 3 min for 10 times while perfusing with 0.1 mM ferrous iron. For iron efflux assay (B), primary cultured astrocytes were perfused with 1 mM ferrous iron for 20 min, then fluorescence intensity was measured every 3 min for 10 times while perfusing with 1 mM DFO, a membrane-impermeant, strong and specific iron chelator, and the intracellular iron was drained out to the medium, indicated by the increase in calcein fluorescence. Data are presented as the mean ± SEM of six independent experiments. n= 6.

  1. Q: 3.4 “Given that hepcidin-FPN1 axis regulates the iron metabolism by feedback effect of hepcidin (ref). : The sentence is unfinished. The references should be added.

A: We apologize for this and thank you for your suggestion. We have rewritten in red written. And added the references.

  1. Q:Figure 5. The text on this figure is blurred, the quality of the figure should be improved.

A: We apologize for this and thank you for your suggestion. We have replaced a new figure.

FIGURE 5. TNF-α and IL-1β mRNA levels in BV2 cells treated with 6-OHDA

  • qRT-PCR assay to measure TNF-α mRNA level. Data are presented as fold changes in TNF-α mRNA expressions in 6-OHDA treatment versus control treated groups. (B) qRT-PCR assay to measure IL-1β mRNA level. Data are presented as fold changes in IL-1β mRNA expressions in 6-OHDA treatment versus control treated groups. Each bar represents the mean±SEM from four independent experiments. n=4. **P< 0.01, compared with the control .

  1. Q:Figure 10. The text at the left upper part is too small, it should be increased. Hepcidin should be placed on a place where it can be seen without interfering with other figure fragments.

A: We apologize for this and thank you for your suggestion. We have replaced a new figure.

FIGURE 10. Graphical illustration of the effect of 6-OHDA on iron metabolism in BV2 cells.

DMT1: divalent metal transporter 1; FPN1: ferroportin 1; IRP1: iron regulatory protein 1; IL-1β: Interleukin-1β; TNF- α : tumor necrosis factor-α.